# Dissipation Dynamics, Terminal Residues and Dietary Risk Assessment of Two Isomers of Dimethacarb in Rice by HPLC-MS/MS

**DOI:** 10.3390/foods10112615

**Published:** 2021-10-28

**Authors:** Shouying Tang, Xiurou Meng, Yongkang Wang, Xueqin Shi, Tianyou Feng, Deyu Hu, Yuping Zhang

**Affiliations:** 1State Key Laboratory Breeding Base of Green Pesticide and Agricultural Bioengineering, Key Laboratory of Green Pesticide and Agricultural Bioengineering, Ministry of Education, Guizhou University, Guiyang 550025, China; tangshouying2021@163.com (S.T.); mxr18285194776@163.com (X.M.); sxq1811995@163.com (X.S.); fty1756833373@163.com (T.F.); gzu_dyhu@126.com (D.H.); 2Clinical Laboratory, Qufu City Center for Disease Control and Prevention, Jining 273100, China; qfsmyqywd@163.com

**Keywords:** dimethacarb, rice, degradation, residues, risk assessment, HPLC-MS/MS

## Abstract

Dimethacarb is a carbamate insecticide developed in China that contains 3,5-dimethylphenyl methylcarbamate (XMC) and 3,4-dimethylphenyl methylcarbamate (MPMC) isomers. Dimethacarb has been registered for use in rice in China, but no residue or degradation of dimethacarb in rice has been reported and the maximum residue limits (MRLs) for rice have not been established. A versatile high-performance liquid chromatography-tandem mass spectrometry (HPLC-MS/MS) method was developed with modified QuEChERS sample preparation to determine two isomers of dimethacarb in rice. The average recovery of XMC and MPMC in brown rice, rice husk, and rice straw ranged from 71.69 to 100.60%, with spike levels of 0.01 to 1 mg/kg and relative standard deviations (RSDs) of 0.21 to 8.41%. Field experiments showed that the half-lives of XMC and MPMC in rice straw were 4.08 to 4.23 days and 3.48 to 3.69 days, respectively. Final residues of XMC and MPMC in rice husk after 21 days of spraying at six sites ranged from 0.23–2.65 mg/kg and 0.06–1.10 mg/kg, and <0.01–0.16 mg/kg and <0.01–0.04 mg/kg in brown rice. The ratio of XMC to MPMC content in the rice husk differed from the original 50% dimethacarb EC, indicating the difference in the degradation rate of XMC and MPMC. The estimated risk quotient (RQ) for both XMC and MPMC was less than 30%. These data for residues from six representative locations could provide a reference for establishing the MRL of dimethacarb in rice.

## 1. Introduction

According to the National Bureau of Statistics, China’s rice acreage was restored to 30.76 million hectares in 2020, an increase of 382,000 hectares compared to 2019. The total rice yield was 211.86 million tons, which means the yield has been stable at more than 200 million tons for 10 consecutive years [1,2,3]. From a food security perspective, rice pests are one of the major factors limiting an increase in rice quality and yield. According to incomplete statistics, the annual loss of rice because of disease and pests in China is about 5 million tons [4]. Rice is affected by a variety of pests during cultivation, including rice planthoppers, *Nilaparvata lugens* (Stal), *Laodelphax striatellus* (Fallen), and *Sogatella furcifera* (Hoeváth), rice borers, *Chilo suppressalis* (Walker), and *Tryporyza incertulas* (Walker), and the rice leaf folder, *Cnaphalocrocis medinalis* Guenee [5,6,7]. In the middle and late period of rice cultivation, the rice planthopper, brown planthopper, and other pests appear with high incidence [8]. Therefore, preventing and controlling rice insect pests during this period will reduce the incidence of insect pests and increase rice yield.

Carbamate insecticides are widely used in food crops, fruits, vegetables, cotton, tobacco, and other commercial crops because of their high insecticidal effect. The insecticidal mechanism of many carbamate pesticides such as dimethacarb, isoprocarb, carbaryl and aldicarb is to inhibit the nerve conduction acetylcholinesterase activity of insects [9,10,11,12]. Dimethacarb (mixed dimethylphenyl-N-methylcarbamate) is a carbamate insecticide made from a mixture of two isomers [3,5-dimethylphenyl methylcarbamate (XMC) and 3,4-dimethylphenyl methylcarbamate (MPMC)], with a molecular weight of 179.2. Figure 1 shows the structural formulas of XMC and MPMC. Dimethacarb has an excellent insecticidal effect on rice planthoppers and it has a high insecticidal rate as most pests fall into the water 1 hour after spraying; additionally, it effectively controls the rice leafhopper and cotton leafhopper [13]. So far, dimethacarb has been approved for use in rice in China. However, China has not set a residue limit for this in rice and there is no literature reporting on the degradation and residue of dimethacarb in rice. In terms of environmental and public safety, it is necessary to determine the dimethacarb in brown rice after its use. 

Given the wide range of uses of carbamate insecticides, the detection of carbamate pesticides in foods is very important. Many studies have established methods of detecting carbamate pesticides in agricultural products such as high-performance liquid chromatography (HPLC), gas chromatography (GC), gas chromatography-tandem mass spectrometry (GC-MS/MS), ultra-performance liquid chromatography-tandem mass spectrometry (UPLC-MS/MS), and high-performance liquid chromatography-tandem mass spectrometry (HPLC-MS/MS) [11,14]. For example, Li et al. applied magnetic graphene solid-phase extraction (SPE) to determine trace carbamate pesticides in tomatoes in combination with HPLC on a Hypersil C18 column [15]. Zhang et al. established a method for simultaneously determining eight types of pesticide residues in carbamate-based pesticides in vegetable samples using GC-MS [16]. Shi et al. used a new adsorbent SPE to determine five carbamate pesticides in environmental water samples by UPLC-MS/MS on a BEH C18 column [17]. Ma et al. established a method for multiple residues analysis of 12 carbamate pesticides in purple cabbage, citrus fruits, watermelons, cucumbers, cowpeas, and lettuce by dispersion solid phase extraction (d-SPE) and UPLC-MS/MS on a BEH C18 column [18]. Atriche et al. determined carbamate pesticides in surface water samples by SPE and LC-MS on a Xterra C18 column [19]. Zhang et al. established a method for simultaneously determining 16 types of carbamate insecticides and 6 types of metabolite residues in egg samples using SPE in combination with LC-MS/MS on a BEH C18 column [20]. However, there is no literature on the residue method for detecting dimethacarb in rice. Besides, dimethacarb contains two isomers. Separation of positional isomers is difficult because of their similar physical and chemical properties [21,22]. The successful separation of the two dimethacarb isomers is the key to further study.

To evaluate the degradation trends of the two dimethacarb isomers in rice and their residue levels in brown rice after application in the field, field experiments were conducted in six major rice-producing provinces in China, and relevant samples were collected from the open field. This study aims to: (1) explore the liquid phase method for separating two isomers in dimethacarb; (2) develop a modified QuEChERS method for simultaneously measuring two dimethacarb isomers residue in brown rice, rice husk, and rice straw by HPLC-MS/MS; (3) study the difference in the dissipation behavior of the two isomers of dimethacarb in rice straw; (4) evaluate the residue levels of XMC and MPMC in brown rice and rice husk; and (5) evaluate the dietary risk of dimethacarb in rice. The results of this study can provide data to support the development of residue limits for dimethacarb in rice.

## 2. Materials and Methods

### 2.1. Chemicals and Reagents

Standards for XMC (99.9% pure) were purchased from Beijing Tan Mo Quality Inspection Technology Co., Ltd. (Beijing, China). MPMC (99.5% pure) was purchased from Chem Service, Inc. (Guangdong, China). 50% dimethacarb EC was purchased from Jiangsu Dongbao Agrochemical Co., Ltd. (Jiangsu, China). Mass spectrometry grade methanol and acetonitrile were purchased from Thermo Fisher Technologies Co., Ltd. (Shanghai, China). Chromatographic grade ethanol was supplied by Honeywell Trading (Shanghai) Co., Ltd. (Shanghai, China). Analytical grade reagents (acetic acid, methanol, acetonitrile, ethyl acetate, dichloromethane) were supplied by Jinshan Chemical Reagent Co. (Chengdu, China). Analytical grade anhydrous Na_2_SO_4_ and NaCl were supplied by Youpu Reagent Company (Tianjin, China). Primary-secondary amine (PSA, 40–63 μm), graphitized carbon black (GCB, 120–400 mesh), and octadecylsilane (C18, 50 μm) were purchased from Agela Technologies (Tianjin, China). The syringe filter (0.22 μm) was obtained from PeakSharp Technologies (Beijing, China).

### 2.2. Standard Solutions

Stock standard solutions of XMC (200 μg/mL) and MPMC (200 μg/mL) in the same volumetric flask were prepared with chromatographic grade methanol. A series of concentrations of standard solutions (0.001, 0.002, 0.005, 0.01, 0.02, 0.05, 0.1, 0.2, 0.5 μg/mL) were obtained by sequentially diluting the stock solutions with methanol. Matrix-matched standards (0.001–0.5 μg/mL) were obtained by evaporating 1 mL of blank sample (brown rice, rice husk, and rice straw) extract at 45 °C and dissolving it in 1 mL of each concentration of standard stock solution. All standard solutions were saved in the dark in a 4 °C refrigerator before use.

### 2.3. Field Trials

Field trials were conducted at six different sites in China: Hunan (subtropical monsoon climate): 113.26° E and 28.28° N), Heilongjiang (cold temperate climate, 126.30° E and 45.81° N), Anhui (warm temperate zone humid monsoon climate, 116.80° E and 33.96° N), Zhejiang (subtropical monsoon climate, 120.68° E and 30.09° N), Guangxi (subtropical monsoon climate, 108.30° E and 22.85° N), Jiangsu (subtropical monsoon climate, 116.67° E and 40.22° N). The field experiment was conducted from July to December 2019. Rice was sprayed with 50% dimethacarb EC before it matured. The formulation was applied two and three times at a low dosage of 750 g a.i./ha (the recommended dose) and at a high dosage of 1125 g a.i./ha (1.5 times the recommended dose). Each experimental plot was 30 m^2^ and replicated three times. At least 1 kg of rice grain samples was randomly collected 7, 14, and 21 days after the last spray, followed by brown rice samples (0.2 kg) and rice husk samples (0.1 kg). For further analysis, the sample was placed in a −20 °C deep freezer.

Field trials, including dissipation experiments were conducted in Hunan (subtropical monsoon climate, 113.26° E and 28.28° N) and Heilongjiang (cold temperate climate, 126.30° E and 45.81° N) to investigate the dissipation dynamics of XMC and MPMC in rice straw. Dimethacarb EC (50%) was sprayed uniformly with a knapsack sprayer at 1125 g a.i./ha (1.5 times the recommended dose), and water was sprayed once on the control. Each experiment was conducted in three 15 m^2^ plots, separated by buffer zones. Plant samples were collected at 2 h, and on the 1st, 3rd, 5th, 7th, 10th, 14th, and 21st day after spraying the pesticide. Rice straw samples were chopped and mixed and stored at −20 °C in a refrigerator until analysis.

### 2.4. Sample Preparation

Rice straw (4 ± 0.02 g), rice husk (2 ± 0.02 g), and brown rice (10 ± 0.02 g) were weighed in a polypropylene centrifuge tube (50 mL), and 1% acetic acid acetonitrile (20 mL) was added. Samples were extracted on a vortex counter at a speed of 2500 rpm for 5 min. Next, 2 g of anhydrous Na_2_SO_4_ and 2 g of NaCl (rice straw, rice husk), 3 g of anhydrous Na_2_SO_4_, and 3 g of NaCl (brown rice) were added, and sonicated for 5 min. Then, the extract was centrifuged (4025× *g*) for 3 min. Next, 1 mL of the supernatant (1% acetic acid acetonitrile phase) was pipetted off and transferred to a 2.5 mL micro-centrifuge tube supplemented with 0.1 g of GCB and the supernatant was vortexed for 30 s. Finally, the purified extracts were filtered through a 0.22 μm nylon membrane filter to analyze with HPLC-MS/MS.

### 2.5. Conditions for the HPLC-MS/MS Analysis

Dimethacarb analysis was performed on an AB Sciex 4500 trap HPLC-MS/MS system with an electrospray ionization source (ESI) (AB Sciex Technologies, Framingham, MA, USA) equipped with an ACQUITY UPLC^®^ BEH Shield RP18 column (100 mm × 2.1 mm, 1.7 µm; Waters, Milford, MA, USA). The mobile phase was acetonitrile/water (30:70, *v/v*) with a flow rate of 0.3 mL/min, a column temperature of 25 °C, and an injection volume of 3 μL. The samples were measured by multiple reaction monitoring (MRM) in positive ion mode. The qualitative ion pair of dimethacarb is 180.1/108.1, the quantitative ion pair is 180.1/123.2, and the collision energy is 49.306 eV. The ion source temperature was 500 °C; the ion sputtering voltage was 5500 V; the curtain gas was 35 psi, and the ion source gas 1 and gas 2 were 55 psi. Under these conditions, the retention times for MPMC and XMC were 7.1 and 7.8 minutes, respectively, the analysis time was 9 minutes and the resolution factor was 1.75.

### 2.6. Validation Method

Two isomers were plotted in nine concentration ranges from 0.001 to 0.5 μg/mL, and one solvent standard curve and three matrix standard curves were generated in terms of concentration and peak area. The limit of quantification (LOQ), limit of detection (LOD), matrix effect (ME), precision and accuracy were validated according to the SANTE/12682/2019 document [23]. The LOD and LOQ of dimethacarb were calculated as the 3:1 signal-to-noise (S/N) ratio and the lowest spike level. To assess the accuracy and precision of the developed method, recovery experiments were performed at three spiked levels (0.01, 0.1, and 1 mg/kg) in blank brown rice, rice husk and rice straw, respectively, using the sample preparation method described above. Five replicates of the spiked samples at three levels of XMC and MPMC for brown rice, rice husk, and rice straw matrix were prepared on three different days. The relative standard deviation (RSD) obtained from repeatability (intra-day) and reproducibility (inter-day) experiments is used to evaluate stability.

In this study, a solvent (1% acetic acid acetonitrile) standard curve and a matrix blank sample extract were used to prepare matrix standard curves to determine the matrix effects (ME). The matrix effect (ME%) was calculated according to Equation (1).
ME(%) = 100 × [S_matrix_ − S_solvent_]/S_solvent_(1)
where ME is the matrix effect and S_matrix_ and S_solvent_ are the slopes of the calibration curves in the matrix and pure solvent, respectively. When |ME| ≤ 20% is considered as a weak or no matrix effect; when 20% < |ME| ≤ 50% is considered as a moderate matrix effect; |ME| > 50% is considered as a strong matrix effect [24]. Positive values of the matrix effect indicate that the analysis response was enhanced and negative values indicate that the analysis response was inhibited by the matrix [25].

Data were analyzed using SPSS 16.0. Differences between individual treatments were analyzed using analysis of variance ANOVA), and the means were compared using Duncan’s new multiple range test. Differences were considered statistically significant at *p* < 0.05.

### 2.7. Dissipation Kinetics of Dimethacarb

The degradation of XMC and MPMC in the three matrices was carried out by plotting the residual concentration as a function of time. The rate constant *k* was calculated by Equation (2), and the half-life (*T*_1/2_) was calculated by Equation (3).
*C_t_* = *C*_0_e^−*kt*^(2)
*T*_1/2_ = ln2/*k* ≈ 0.693/*k*(3)
where *C*_0_ represents the initial concentration of XMC and MPMC (mg/kg), *C_t_* represents the concentration of XMC and MPMC (mg/kg) at time *t* (day), and *k* is the degradation rate constant.

### 2.8. Dietary Risk Assessment

The long-term dietary risk assessments of the two isomers of dimethacarb in brown rice were calculated using Equations (4) and (5).
NEDI = ∑(STMRi × *Fi*)(4)
RQ = NEDI/(ADI × bw) × 100%(5)
where NEDI (mg) is estimated daily intake, STMRi (mg/kg) is the median XMC and MPMC residue in China’s brown rice, *Fi* (kg) is national consumption of a specific agricultural or food product (kg/d bw), acceptable daily intake (ADI) is based on mg/kg body weight, bw is the average weight of a Chinese adult (63 kg), and RQ is the risk quotient. When the RQ value is higher, it represents higher pesticide residues, an RQ > 100% represents unacceptable risk, while an RQ < 100% represents acceptable risk [26,27].

## 3. Results and Discussion

### 3.1. Method Development

The QuEChERS method uses an adsorbent filler to interact with impurities in the matrix, adsorbing the impurities, removing the impurities, and purifying the object to be measured. This is a quick, easy, economical, efficient, interference-free, and safe pretreatment method. This article aimed to establish an improved QuEChERS method for the extraction and detection of XMC and MPMC in brown rice, rice husk, and rice straw.

#### 3.1.1. Extraction of Brown Rice, Rice Husk, and Rice Straw

The choice of extraction solvent is the key to the residual analysis method. In this study, five organic solvents (methanol, acetonitrile, ethyl acetate, dichloromethane, and 1% acetic acid acetonitrile, 20 mL each) were selected as extraction solvents and to demonstrate the extraction efficiency of XMC and MPMC in brown rice, rice husk, and rice straw. Rice was studied at an addition level of 0.1 mg/kg (Figure 2).

Of the five solvents, ethyl acetate and dichloromethane were not effective in extracting XMC and MPMC from brown rice, rice husk, and rice straw. Some reagents were suitable for the extraction of rice straw, but not for the extraction of brown rice. For example, the extraction recoveries of XMC and MPMC in methanol extraction of brown rice were only 52.64% and 53.16%. Some reagents had a good extraction effect on XMC, but had an inadequate extraction effect on MPMC. For example, acetonitrile had an extraction efficiency of only 66.00% in the MPMC of rice straw. The extraction efficiency of XMC and MPMC from brown rice, rice husk, and rice straw with 1% acetic acid acetonitrile was 84.10% to 101.12%. Therefore, 1% acetic acid acetonitrile provided the highest extraction rate for XMC and MPMC.

#### 3.1.2. Clean-Up for Brown Rice, Rice Husk, and Rice Straw

In this work, we analyzed the interference of three cleaning agents (PSA, C18, and GCB) in the recovery of the three matrices (Figure 3). C18 removes polar compounds, PSA absorbs polar compounds, and GCB absorbs pigments and sterols. In many experiments, GCB has reduced the recovery rate of the target pesticide and C18 and PSA have been selected as the purifying agents [28,29,30]. However, GCB was able to effectively remove impurities from the three matrix samples without decreasing the recovery. In purifying brown rice, rice husk, and rice straw matrix, the recovery rates of XMC and MPMC using C18 (100 mg) as the cleaning agent were 76.42 to 93.89% and 74.37 to 90.43%, respectively, while the rates for MPMC using GCB (100 mg) as a cleaning agent were 89.48–101.16% and 84.10–96.44%, respectively. Both C18 and GCB resulted in significantly higher recoveries than PSA for the two target compounds in rice straw (*p* < 0.05), and the recoveries using C18 and GCB were closer to 100%. The recoveries of the two compounds cleaned with GCB in rice husk were closest to 100%. Both C18 and GCB had significantly lower recoveries than PSA for the two compounds in rice brown rice (*p* < 0.05). However, the recoveries with C18 and GCB were closer to 100% (Figure 3). When using GCB, the extract became almost colorless, showing the strongest removal of impurities. Therefore, GCB was used as the purifying agent in the purification process. 

### 3.2. Validation Method

Matrix-matched calibration curves were plotted for nine concentrations (0.001, 0.002, 0.005, 0.01, 0.02, 0.05, 0.1, 0.2 and 0.5 μg/mL) of XMC and MPMC in standard solutions and matrix standard solutions (brown rice, rice husk, and rice straw) with correlation coefficients (*R*^2^) of 0.9981 to 0.9998, as shown in Table 1. The data in Table 1 showed that the brown rice samples had a slightly improved response to XMC and MPMC, with MEs of 9.1% and 8.0%, respectively. However, XMC and MPMC were slightly inhibited in the rice straw and rice husk samples, respectively. The ME of XMC and MPMC in rice straw was −3.2% and −4.0%, respectively. The ME of XMC and MPMC in the rice husk was −3.6% and −3.7%, respectively. XMC and MPMC were weak matrix effects in all three matrices, reflecting the good purifying effect of GCB on the rice samples. In our study, the value of the LODs was calculated based on the signal-to-noise ratio of 3:1, and the LOQs were set as the lowest spiked level. LOQs of XMC and MPMC in brown rice, rice husk, and rice straw were 0.01 mg/kg. The LODs were estimated at 0.0005 mg/kg for XMC and MPMC in rice straw, 0.0009 mg/kg in rice husk and 0.0002 mg/kg in brown rice (Table 1). 

Subsequently, a recovery study was developed using five consecutive extracts at three spike levels (0.01, 0.10, 1.00 mg/kg) for brown rice, rice husk, and rice straw. The mean recoveries of XMC and MPMC were 75.26–98.98% and 71.69–100.60% with RSDs ≤ 5.59% and ≤8.41%, respectively, and the inter-day RSDs (*n* = 15) of the method were 2.40–7.19% (Table 2). Therefore, the recoveries of the two isomers at three different spiked levels were satisfactory in the three samples, and the developed analytical method meets the performance and reproducibility requirements specified in the SANTE/11813/guidelines [31]. Typical HPLC-MS/MS chromatograms of XMC and MPMC are shown in Figure 4. 

### 3.3. Dissipation of Two Isomers of Dimethacarb in Two Sites

Samples of rice straw were collected after spraying at 2 h, 1, 3, 5, 7, 10, 14 and 21 days, and the XMC and MPMC residues in the samples were analyzed. According to the plotted dynamic dissipation curves, the kinetic equations of XMC in rice straw was *C_t_* = 0.4470e^−0.170*t*^ (Hunan, *R*^2^ = 0.91) and *C_t_* = 0.3864e^−0.164*t*^ (Heilongjiang, *R*^2^ = 0.74), with half-lives of 4.08 days and 4.23 days, respectively, and the kinetic equation of MPMC in rice straw was *C_t_* = 0.1414e^−0.199*t*^ (Hunan, *R*^2^ = 0.93) and *C_t_* = 0.0861e^−0.188*t*^ (Heilongjiang, *R*^2^ = 0.71), with half-lives of 3.48 days and 3.69 days, respectively. Figure 5 shows the dissipation curves of XMC and MPMC in rice straw. The results indicated that the degradation of XMC and MPMC in rice straw conformed to the first-order kinetic model (Equation (2)). Many experimental results have shown that the rate of pesticide dissipation in plants is affected by different regions and different environments. Our results show that the half-lives of dimethacarb in rice straw in Heilongjiang and Hunan happen to be similar and dimethacarb dissipated rapidly in rice straw (half-life < 5 days). However, the degradation rate of MPMC in rice straw was slightly faster than that of XMC. 

### 3.4. Terminal Residues

Dimethacarb EC (50%) was sprayed evenly from 750 g a.i./ha (the recommended dose) to 1125 g a.i./ha (1.5 times the recommended dose) 2–3 times at 7-day intervals. Table 3 and Table 4 show the terminal residues of dimethacarb in rice husk and brown rice at Hunan, Heilongjiang, Anhui, Zhejiang, Guangxi, and Jiangsu. Our six trial sites cover the major rice producing areas of China.

#### 3.4.1. Terminal Residue of Dimethacarb on the Rice Husk

The residues of XMC in rice husk at 7, 14, and 21 days after spraying at 6 sites were 1.18–4.70 mg/kg, 0.60–3.78 mg/kg, 0.23–2.65 mg/kg, and MPMC in rice husk were 0.38–2.12 mg/kg, 0.17–1.70 mg/kg, 0.06–1.10 mg/kg, respectively (Table 3). We observed that the terminal residue of MPMC was less than that of XMC, mainly because the content of XMC was 2.9 times that of MPMC in 50% dimethacarb EC. At 7, 14, and 21 days after spraying, the ratio of XMC to MPMC in most samples differed from the value of 2.9 in the original preparation. The minimum ratio of XMC to MPMC was 2.22 on the 7th day in Heilongjiang, and the maximum was 5.14 on the 21st day in Guangxi. The harvest ratio was greater than 2.9, and the degradation rate of MPMC on rice husk was faster than the rate of XMC degradation, presumably leading to the enrichment of XMC in the sample and vice versa. The harvest ratio is less than 2.9, indicating that the degradation rate of MPMC on rice husk was slower than that of XMC.

The terminal residues of XMC and MPMC in rice husk in Heilongjiang were higher than those in the other five regions, at 1.25–4.7 mg/kg and 0.48–2.12 mg/kg, respectively. The terminal residues of XMC and MPMC in the rice husk of Guangxi were all lower than those of the other five regions, 0.51 to 1.70 mg/kg and 0.12 to 0.48 mg/kg, respectively. Crop varieties, sunlight, ambient temperature, crop growth, initial pesticide concentrations, wind speed, rainfall, etc., affect field residues. We compared several environmental factors, such as sunlight, temperature, wind speed, and rainfall in six locations. The rice varieties in the six rice-producing areas are different: Hunan (Xiangyou 269), Heilongjiang (Daohuaxiang), Anhui (Xuyou 733), Zhejiang (Nanjing 46), Guangxi (Y liangyou 2), and Jiangsu (Zhendao 18). In field experiments in Guangxi, 13 to 16 rainfalls were observed, and the weather was mostly cloudy or overcast with high humidity. In the other five locations, there were less than 13 times of rainfall and sporadic rainfall. The field trials in Heilongjiang were conducted with little rainfall and low wind speed.

#### 3.4.2. Terminal Residue of Two Isomers of Dimethacarb on the Brown Rice

The XMC and MPMC residues of brown rice were lower than those of the six rice husks. Residues of XMC and MPMC in brown rice at 7, 14, and 21 days after spraying at the six locations were <0.01–0.28 mg/kg and <0.01–0.09 mg/kg, respectively (Table 4). Twenty-one days after spraying in six locations, brown rice XMC and MPMC residues were <0.01–0.16 mg/kg and <0.01–0.04 mg/kg, respectively. The terminal residues of brown rice XMC and MPMC at the six experimental sites were very low. Also, there was no significant difference in the terminal residue of brown rice between Heilongjiang, which has a large terminal residue of rice husks, and Guangxi Zhuang, which has a small terminal residue. The results showed that rice husk blocked most pesticides on brown rice, suggesting that dimethacarb may not be a systemic pesticide for rice. This result could provide the basis for relevant departments in China to establish the MRL of dimethacarb in rice. The end-of-life residual data from this study will help assess the risk of dimethacarb in brown rice.

### 3.5. Dietary Intake Risk Assessment

Currently, the Ministry of Agriculture of China does not provide ADI values for dimethacarb. The ADI values for carbamate insecticides range from 0.001–0.03 mg/kg bw [32]. The smallest ADI value is 0.001 mg/kg bw carbofuran. Therefore, on the basis of the highest risk assessment, the ADI of XMC and MPMC was set to 0.001 mg/kg bw for calculation by analogy [33]. Dimethacarb was registered only in rice. According to the Chinese Dietary Guidelines (the 2011 revision), the daily intake of rice and its products in the Chinese dietary pattern is 0.2399 kg per person [34]. The median XMC residues at 7, 14, and 21 days after spraying were 0.07 mg/kg, 0.04 mg/kg, and 0.03 mg/kg, respectively, and the median MPMC residues were 0.03 mg/kg, 0.02 mg/kg, and < 0.01 mg/kg, respectively. To assess maximum risk, we selected and assessed the residual median of 7 days. The estimated daily intakes (NEDI) for XMC and MPMC from supervised residue trials data were 0.0168 mg and 0.0072 mg, respectively. The RQ values for XMC and MPMC were 26.7% and 11.4%, respectively. When the formulation was applied two and three times at dosages from 750–1125 g a.i./ha at an interval of 7 days, and the rice grain samples was collected 7 days after the last spray, they did not show obvious dietary risk.

## 4. Conclusions

In this study, a rapid and versatile HPLC-MS/MS method was established with modified QuEChERS sample preparation to determine two isomers of dimethacarb in rice. The mean recoveries were 71.69–100.60% with RSDs ≤ 8.41%. The results showed that XMC and MPMC rapidly dissipated in rice straw (half-life < 5 days). Residues of XMC and MPMC in the rice husk at 7, 14, and 21 days after spraying from six sites were 0.23 to 4.70 mg/kg and 0.06 to 2.12 mg/kg, respectively. The ratio of XMC to MPMC in the rice husk sample at harvest differed from the 50% dimethacarb EC preparation. Twenty-one days after spraying at six sites, residues of XMC and MPMC in brown rice were <0.01–0.16 mg/kg and <0.01–0.04 mg/kg, respectively. Presently, China has not established MRL standards for XMC and MPMC in brown rice. Risk assessment for the development of MRL standards needs residue data from a sufficiently large and fully representative number of experimental sites. In this study, our six experimental sites covered the major rice production areas in China. The results of this study provide a basis for the establishment of MRLs for XMC and MPMC in rice.

## Figures and Tables

**Figure 1 foods-10-02615-f001:**
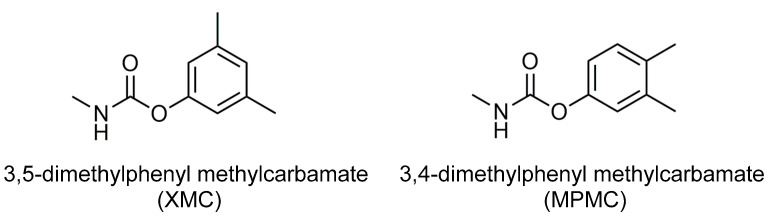
Chemical structures of dimethacarb.

**Figure 2 foods-10-02615-f002:**
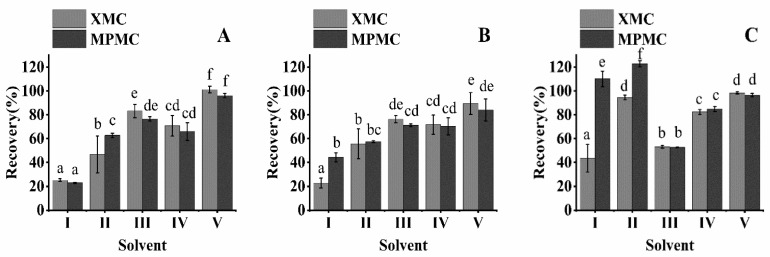
Effect of different solvents on dimethacarb extraction in three matrix samples (spike at 0.1 mg/kg): (**A**) rice straw, (**B**) rice husk, and (**C**) brown rice. (Ⅰ) ethyl acetate, (Ⅱ) dichloromethane, (Ⅲ) methanol, (Ⅳ) acetonitrile, (Ⅴ) 1% acetic acid acetonitrile. (abcdef: different letters represent statistically significant differences between the extraction effects of different extractants on XMC and MPMC, *p* < 0.05).

**Figure 3 foods-10-02615-f003:**
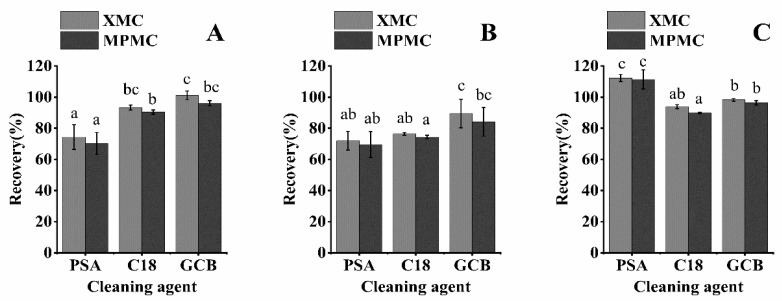
Effect of various cleaning agents on the purification of three matrix samples (spike at 0.1 mg/kg): (**A**) rice straw, (**B**) rice husk, and (**C**) brown rice. (abc: different letters represent statistically significant differences between the recovery rates of XMC and MPMC with different cleaning agents, *p* < 0.05).

**Figure 4 foods-10-02615-f004:**
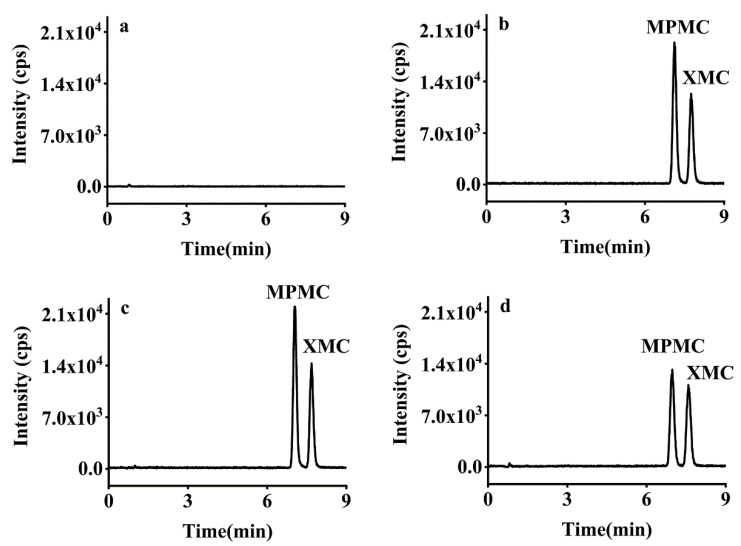
Typical HPLC-MS/MS chromatograms of dimethacarb in (**a**) blank brown rice (**b**) solvent standard solution (0.005 μg/mL), (**c**) matrix-matched standard solution (0.005 μg/mL), and (**d**) spiked brown rice (0.01 mg/kg).

**Figure 5 foods-10-02615-f005:**
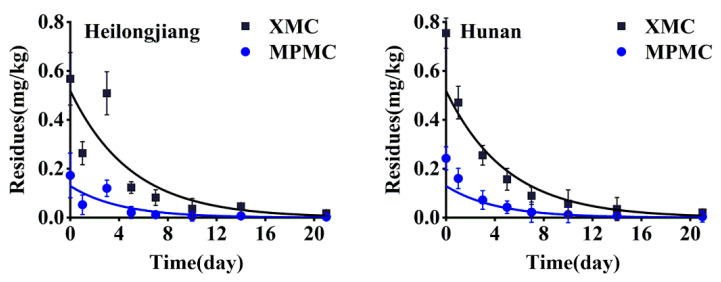
The dissipative curve of dimethacarb in rice straw.

**Table 1 foods-10-02615-t001:** Standard curve equation of dimethacarb.

Matrices	Concentrations (μg/mL)	Analytes	Linear Equation	*R* ^2^	LOQs (mg/kg)	LODs (mg/kg)	ME(%)
Methanol	0.001–0.5	XMC	y = 28910435x + 85887	0.9986	/	/	/
		MPMC	y = 32641586x + 126652	0.9981	/	/	/
Rice Straw	0.001–0.5	XMC	y = 27987162x + 71944	0.9998	0.01	0.0005	−3.2%
		MPMC	y = 31346715x + 89803	0.9998	0.01	0.0005	−4.0%
Rice Husk	0.001–0.5	XMC	y = 27864126x + 62260	0.9995	0.01	0.0009	−3.6%
		MPMC	y = 31432383x + 94868	0.9992	0.01	0.0009	−3.7%
Brown Rice	0.001–0.5	XMC	y = 31532250x + 110620	0.9988	0.01	0.0002	9.1%
		MPMC	y = 35269206x + 141208	0.999	0.01	0.0002	8.0%

Note: LOQ: limit of quantification; LOD: limit of detection; ME: matrix effect.

**Table 2 foods-10-02615-t002:** Recoveries and RSDs for dimethacarb in rice.

Analytes	Matrices	Spiked Levels (mg/kg)	Average Recovery (%), Intraday RSD (%, *n* = 5)						Interday RSD (%, *n* = 15)
			Day 1		Day 2		Day 3		
XMC	Rice Straw	0.01	89.68	4.67	89.47	3.53	93.77	2.86	4.13
		0.1	75.26	4.35	79.10	2.83	77.33	2.06	3.63
		1	75.88	5.05	76.83	1.23	78.15	2.19	3.24
	Rice Husk	0.01	91.33	5.21	87.93	5.48	88.63	5.59	6.09
		0.1	98.73	2.85	98.98	2.70	94.01	1.58	3.34
		1	77.88	2.23	76.70	3.13	81.04	0.91	3.19
	Brown Rice	0.01	80.32	1.61	81.17	4.90	82.47	3.66	3.57
		0.1	88.43	1.19	95.15	2.63	87.37	2.23	4.36
		1	81.63	3.18	80.47	5.19	86.82	5.08	5.48
MPMC	Rice Straw	0.01	82.88	1.54	80.19	1.51	86.74	1.33	3.60
		0.1	74.71	3.55	78.76	2.77	77.39	0.75	3.31
		1	74.54	4.66	71.69	0.21	72.92	1.66	3.16
	Rice Husk	0.01	94.24	5.48	92.65	8.41	86.61	5.47	7.19
		0.1	100.46	1.95	100.60	2.23	93.85	0.73	3.71
		1	84.37	1.58	83.07	3.47	76.89	1.26	4.68
	Brown Rice	0.01	73.09	2.04	74.43	2.63	72.76	2.30	2.40
		0.1	83.05	1.13	90.39	2.28	84.22	2.85	4.39
		1	76.61	2.12	75.93	3.48	82.00	4.71	4.94

Note: RSD: relative standard deviation.

**Table 3 foods-10-02615-t003:** Field-incurred residues of dimethacarb in rice husk.

Sites	Dosage (g a.i./ha)	Spraying Times	Days after Application	Rice Husk (Mean, mg/kg)		Multiple ^a^	Dosage (g a.i./ha)	Spraying Times	Days after Application	Rice Husk (Mean, mg/kg)		Multiple ^a^
				XMC	MPMC					XMC	MPMC	
Hunan	750	2	7	1.40 ± 0.07	0.45 ± 0.03	3.11	1125	2	7	2.64 ± 0.10	0.91 ± 0.03	2.90
			14	0.72 ± 0.05	0.20 ± 0.02	3.60			14	1.28 ± 0.11	0.38 ± 0.04	3.37
			21	0.69 ± 0.03	0.19 ± 0.01	3.63			21	0.76 ± 0.04	0.22 ± 0.02	3.45
		3	7	1.82 ± 0.13	0.63 ± 0.14	2.89		3	7	3.64 ± 0.27	1.13 ± 0.15	3.22
			14	0.65 ± 0.02	0.17 ± 0.01	3.82			14	1.37 ± 0.09	0.42 ± 0.02	3.26
			21	0.61 ± 0.07	0.15 ± 0.01	4.07			21	1.03 ± 0.05	0.31 ± 0.01	3.32
Heilongjiang	750	2	7	2.84 ± 0.35	1.25 ± 0.16	2.27	1125	2	7	4.30 ± 0.43	1.89 ± 0.21	2.28
			14	2.28 ± 0.24	0.95 ± 0.11	2.40			14	2.37 ± 0.35	1.01 ± 0.15	2.35
			21	1.25 ± 0.08	0.48 ± 0.03	2.60			21	1.38 ± 0.11	0.56 ± 0.07	2.46
		3	7	2.70 ± 0.14	1.18 ± 0.06	2.29		3	7	4.70 ± 0.30	2.12 ± 0.13	2.22
			14	2.01 ± 0.25	0.85 ± 0.11	2.36			14	3.78 ± 0.27	1.70 ± 0.12	2.22
			21	1.55 ± 0.12	0.60 ± 0.05	2.58			21	2.65 ± 0.11	1.08 ± 0.06	2.45
Zhejiang	750	2	7	1.18 ± 0.03	0.51 ± 0.03	2.31	1125	2	7	2.13 ± 0.15	0.92 ± 0.06	2.32
			14	0.60 ± 0.02	0.22 ± 0.00	2.73			14	1.16 ± 0.07	0.44 ± 0.04	2.64
			21	0.24 ± 0.01	0.07 ± 0.00	3.43			21	0.47 ± 0.04	0.12 ± 0.01	3.92
		3	7	1.39 ± 0.08	0.56 ± 0.03	2.48		3	7	2.52 ± 0.14	1.00 ± 0.05	2.52
			14	0.66 ± 0.05	0.24 ± 0.02	2.75			14	1.30 ± 0.08	0.42 ± 0.02	3.10
			21	0.23 ± 0.02	0.06 ± 0.01	3.83			21	1.34 ± 0.02	0.45 ± 0.01	2.98
Guangxi	750	2	7	1.23 ± 0.13	0.43 ± 0.04	2.86	1125	2	7	1.46 ± 0.09	0.48 ± 0.05	3.04
			14	0.84 ± 0.09	0.21 ± 0.02	4.00			14	0.88 ± 0.08	0.22 ± 0.05	4.00
			21	0.51 ± 0.11	0.12 ± 0.04	4.25			21	0.86 ± 0.04	0.18 ± 0.01	4.78
		3	7	1.27 ± 0.18	0.38 ± 0.08	3.34		3	7	1.70 ± 0.03	0.48 ± 0.00	3.54
			14	0.75 ± 0.02	0.17 ± 0.00	4.41			14	1.06 ± 0.18	0.27 ± 0.06	3.93
			21	0.72 ± 0.05	0.14 ± 0.01	5.14			21	1.03 ± 0.05	0.21 ± 0.02	4.90
Jiangsu	750	2	7	1.68 ± 0.15	0.70 ± 0.07	2.40	1125	2	7	2.87 ± 0.19	1.20 ± 0.08	2.39
			14	2.02 ± 0.12	0.80 ± 0.04	2.53			14	2.28 ± 0.11	0.89 ± 0.06	2.56
			21	1.33 ± 0.13	0.44 ± 0.04	3.02			21	1.23 ± 0.02	0.42 ± 0.01	2.93
		3	7	3.37 ± 0.35	1.40 ± 0.16	2.41		3	7	4.03 ± 0.08	1.67 ± 0.01	2.41
			14	2.44 ± 0.17	0.90 ± 0.07	2.71			14	2.68 ± 0.15	1.02 ± 0.08	2.63
			21	0.98 ± 0.04	0.32 ± 0.01	3.06			21	2.23 ± 0.14	0.75 ± 0.04	2.97
Anhui	750	2	7	1.71 ± 0.07	0.58 ± 0.03	2.97	1125	2	7	2.96 ± 0.19	1.11 ± 0.08	2.66
			14	1.20 ± 0.06	0.49 ± 0.02	2.46			14	2.13 ± 0.15	0.92 ± 0.05	2.30
			21	0.91 ± 0.05	0.31 ± 0.02	2.95			21	1.58 ± 0.13	0.59 ± 0.05	2.65
		3	7	1.60 ± 0.21	0.58 ± 0.08	2.79		3	7	2.91 ± 0.16	1.14 ± 0.04	2.56
			14	1.39 ± 0.05	0.56 ± 0.02	2.50			14	3.29 ± 0.48	1.33 ± 0.17	2.48
			21	1.33 ± 0.01	0.48 ± 0.00	2.78			21	2.61 ± 0.14	1.10 ± 0.05	2.37

Note: ^a^ The ratio of XMC to MPMC content in the rice husk.

**Table 4 foods-10-02615-t004:** Field-incurred residues of dimethacarb in brown rice.

Sites	Dosage (g a.i./ha)	Spraying Times	Days after Application	Brown Rice (Mean, mg/kg)		Dosage (g a.i./ha)	Spraying Times	Days after Application	Brown Rice (Mean, mg/kg)	
				XMC	MPMC				XMC	MPMC
Hunan	750	2	7	0.05	0.02	1125	2	7	0.10	0.05
			14	0.01	<0.01			14	0.04	0.02
			21	0.01	<0.01			21	0.02	0.01
		3	7	0.06	0.02		3	7	0.15	0.07
			14	0.02	<0.01			14	0.04	0.02
			21	<0.01	<0.01			21	0.02	0.01
Heilongjiang	750	2	7	0.06	0.02	1125	2	7	0.08	0.03
			14	0.04	0.01			14	0.06	0.02
			21	0.03	<0.01			21	0.03	<0.01
		3	7	0.05	0.02		3	7	0.09	0.03
			14	0.04	0.01			14	0.08	0.03
			21	0.04	0.01			21	0.05	0.02
Zhejiang	750	2	7	0.04	0.01	1125	2	7	0.10	0.03
			14	0.03	<0.01			14	0.05	<0.01
			21	0.01	<0.01			21	0.02	<0.01
		3	7	0.06	0.01		3	7	0.18	0.06
			14	0.03	<0.01			14	0.12	0.03
			21	<0.01	<0.01			21	0.06	0.01
Guangxi	750	2	7	0.04	0.01	1125	2	7	0.06	0.03
			14	0.02	<0.01			14	0.04	0.02
			21	0.01	<0.01			21	0.04	0.02
		3	7	0.05	0.01		3	7	0.09	0.03
			14	0.02	<0.01			14	0.06	0.03
			21	0.02	<0.01			21	0.04	0.02
Jiangsu	750	2	7	0.08	0.03	1125	2	7	0.14	0.05
			14	0.11	0.04			14	0.12	0.04
			21	0.06	0.01			21	0.07	0.02
		3	7	0.17	0.06		3	7	0.28	0.09
			14	0.15	0.05			14	0.19	0.06
			21	0.04	<0.01			21	0.16	0.04
Anhui	750	2	7	0.04	0.02	1125	2	7	0.09	0.04
			14	0.03	<0.01			14	0.05	0.02
			21	0.01	<0.01			21	0.04	0.01
		3	7	0.05	0.02		3	7	0.10	0.05
			14	0.03	0.01			14	0.09	0.04
			21	0.02	<0.01			21	0.07	0.03

## Data Availability

Data are contained within the article.

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
