# Peer review of "Dissipation Dynamics, Terminal Residues and Dietary Risk Assessment of Two Isomers of Dimethacarb in Rice by HPLC-MS/MS"

_foods, 2021, doi:10.3390/foods10112615_

Round 1

Reviewer 1 Report

This paper reported dissipation and risk assessment of dimethacarb in rice with HPLC-MSMS. The content is very important to the related field. There are some points to be revised.  

The ADI of dimethacarb is not established yet. But, authors assessed risk of this compound  with the ADI of isoprocarb. Although the structure of isoprocarb is similar with dimethacarb, toxicity is not related with the structure. I recommend to use the lowest ADI from cabarmate insecticides for food safety, and rewirte this part. 

Minor corrections are as follow.

2 page  line 4, isoprocorb => isoprocarb

3page, delete 3,5-dimethylphenyl methylcarbamate and 3,4-dimethylphenyl methylcarbamate. These are already explained in the previous paragraph. 

3page , in "Field trials" section. plot size is 30 mnot 30 m2

4 page, line 6. pending analysis => until analysis

4 page, line 12. 1 ml => 1 mL

4 page, line 15. filter membrane to HPLC-MS/MS for analysis => mebrane filter to analyze with HPLC-MS/MS.

5 page, formula 3.  lin2/kt => ln2/k

Reviewer 2 Report

The manuscript entitled "Dynamics of Dissipation, Terminal Residues and Dietary Risk Assessment of Two Isomers of Dimethacarb in Rice by HPLC-MS / MS" reports a large study carried out with rice and its plant parts, planted in China.

Contribute with information relevant to sanitary control that serve as parameters for acceptance of the concentration levels of pesticides used.

The authors performed the necessary tests for the conclusions proposed in the manuscript. Analytical rigor in the revealed data is observed. 

Reviewer 3 Report

It is opinion of the reviewer that this paper needs several modifications/corrections. My individual comments are listed below.

The title – It should be “… Dimethylcarbamate …”.

Only one reference ([32]) was cited in Discussion.

47 – It should be “… pesticides such as dimethcarb … is to inhibit …”.

Fig. 3 & 3 – Significance of differences must be marked with letters.

Fig. 3 – The unit of “Intensity” should be added to description of the Y axis.

65/66 – It should be “high performance ….high performance”.

65/68 – What type of the HPLC columns were used?

89 – It should be “dimethylcarbone”.

147 – The centrifugation must be characterized by „x g” instead of „rpm”.

152 – It should be “Condition of an HPLC0MS/MS analysis”.

163 – It should be “resolution factor”.

188 – It should be “The degradation … was carried out.

208 – It should be “This article aimed …”.

211 – It should be “… a key …”.

231/233 – It should be In this work …(Fig. 3). C18 removes ….sterols.”.

231 and other lines – It should be “C18”.

The section of statistical analysis must be added to Methods.

Few abbreviations of journal titles must be corrected.

Round 2

Reviewer 3 Report

The authors corrected this paper properly taken under considerations all my comments. Therefore, I can accept it now.